



# Impact of biogenic SOA loading on the molecular composition of wintertime PM2.5 in urban Tianjin: an insight from Fourier transform ion cyclotron resonance mass spectrometry

Shujun Zhong[1], Shuang Chen[1], Junjun Deng[1], Yanbing Fan[1], Qiang Zhang[1], Qiaorong Xie[1], Yulin Qi[1], Wei Hu[1], Libin Wu[1], Xiaodong Li[1], Chandra Mouli Pavuluri[1], Jialei Zhu[1], Xin Wang[1], Di Liu[2], Xiaole Pan[2], Yele Sun[2], Zifa Wang[2], Yisheng Xu[3], Haijie Tong[4], Hang Su[5], Yafang Cheng[5], Kimitaka Kawamura[6], and Pingqing Fu[1,*]

[1]Institute of Surface-Earth System Science, School of Earth System Science, Tianjin University, Tianjin, China
[2]LAPC, Institute of Atmospheric Physics, Chinese Academy of Sciences, Beijing, China
[3]State Key Laboratory of Environmental Criteria and Risk Assessment, Chinese Research Academy of Environmental Sciences, Beijing 100012, China
[4]Institute of Surface Science, Helmholtz-Zentrum Hereon, 21502 Geesthacht, Germany
[5]Max Planck Institute for Chemistry, Mainz, Germany
[6]Chubu Institute for Advanced Studies, Chubu University, Kasugai, Japan

*Correspondence to*: Pingqing Fu (fupingqing@tju.edu.cn)

**Abstract.** Biomass burning is one of the key sources of urban aerosols in the North China Plain, especially in winter when the impact of secondary organic aerosols (SOA) formed from biogenic volatile organic compounds (BVOCs) is generally considered to be minor. However, little is known about the influence of biogenic SOA loading on the molecular composition of wintertime organic aerosols. Here, we investigated the water-soluble organic compounds in fine particles (PM2.5) from urban Tianjin by ultrahigh-resolution Fourier transform ion cyclotron resonance mass spectrometry (FT-ICR MS). Our results show that most of the CHO and CHON compounds were derived from biomass burning; they contain O-poor and highly unsaturated compounds with aromatic rings, which are sensitive to photochemical reactions, and some of which probably contribute to light-absorbing chromophores. Under moderate to high SOA loading conditions, the nocturnal chemistry is more efficient than photooxidation to generate secondary CHO and CHON compounds with high oxygen content. Under low SOA-loading, secondary CHO and CHON compounds with low oxygen content are mainly formed by photochemistry. Secondary CHO compounds are mainly derived from oxidation of monoterpenes. But nocturnal chemistry may be more productive to sesquiterpene-derived CHON compounds. In contrast, the number- and intensity-weight of S-containing groups (CHOS and CHONS) increased significantly with the increase of biogenic SOA-loading, which agrees with the fact that a majority of the S-containing groups are identified as organosulfates and nitrooxy-organosulfates that are derived from the oxidation of BVOCs. Terpenes may be potential major contributors to the chemical diversity of organosulfates and nitrooxy-organosulfates under photo-oxidation. While the nocturnal chemistry is more beneficial to the formation of organosulfates and nitrooxy-organosulfates under low SOA-loading. The SOA-loading is an important factor associating with the oxidation degree, nitrate



group content and chemodiversity of nitrooxy-organosulfates. Furthermore, our study suggests that the hydrolysis of nitrooxy-organosulfates is a possible pathway for the formation of organosulfates.

# 1 Introduction

Organic aerosols account for about 20-90% of fine particulate matter ($PM_{2.5}$) in the ambient air (Kleindienst et al., 2007;Goldstein and Galbally, 2007;Fan et al., 2016). Various emission sources, including industrial production, coal combustion, traffic emissions, cooking, and biomass burning, influence the formation of haze pollution in the North China Plain (NCP) (Sun et al., 2013). Among these emission sources, biomass burning plays a prominent role in air quality and climate change. Biomass burning emissions include high concentrations of primary organic aerosol (POA), semivolatile organic compounds (SVOCs), and volatile organic compounds (VOCs) (Koss et al., 2018;Engling et al., 2013;Andreae and Merlet, 2001). Humic-like substances (HULIS) account for 28-34% of organic carbon (OC) in the $PM_{2.5}$ emitted by incomplete combustion of solid biomass fuels in domestic stoves (Lin et al., 2010;Park and Yu, 2016). Domestic biomass combustion is also one of the key emission sources of HULIS, especially in winter and spring in the North China Plain (Li et al., 2019). Previous studies have revealed that biomass burning particles contain plenty of important chromophores such as nitroaromatics and N-heterocyclic compounds, which can enhance the light absorption of aerosols (Wang et al., 2019). A recent study revealed that most of light absorption of brown carbon (BrC) is related to biomass burning particles, with minor amount from biogenic SOA (Washenfelder et al., 2015). There are significant high columnar light absorbing levels in areas with frequent biomass burning activities (Arola et al., 2011). Therefore, biomass burning has been identified as an important contributor to atmospheric BrC (Yan et al., 2020;Saleh et al., 2014;Lin et al., 2016;Yue et al., 2022). Furthermore, large amounts of isoprene, toluene, propylene, and $O_3$ precursors have been measured with high total ozone formation potential values from biomass burning activities such as wheat straw burning in the northern China (Zhu et al., 2016;Fu et al., 2012).

Secondary organic aerosols (SOA) are main products of the oxidation of volatile organic compounds (VOCs) in the presence of OH and $NO_3$ radicals, $O_3$, and other oxidants (Hallquist et al., 2009;Nie et al., 2022;Gentner et al., 2017). Organosulfates (OSs, $ROS(O)_2OH$, esters of sulfuric acid) have been found as the most abundant organosulfur compounds in atmospheric particulate matter, contribute significantly to the mass of SOA and play an important role in their formation pathways (Tolocka and Turpin, 2012;Brüggemann et al., 2020;Surratt et al., 2008;Fan et al., 2022). Moreover, nitrooxy-organosulfates (nitrooxy-OSs), which contain both nitrooxy ($-ONO_2$) and sulfate ester group ($-OSO_3H$), also contribute greatly to the formation of SOA (Brüggemann et al., 2020;Surratt et al., 2008;Xie et al., 2022). Previous studies have shown that both organosulfates and nitrooxy-organosulfates are mostly assumed to be formed by multiphase reactions between acidic sulfate particles and organic compounds from both biogenic and anthropogenic sources (Surratt et al., 2008;Iinuma et al., 2007;Zhang et al., 2012;Kristensen and Glasius, 2011;Riva et al., 2016;Fan et al., 2022). Long aliphatic chain organosulfates and nitrooxy-organosulfates ($C_{17-28}$) are characterized by low degrees of oxidation and unsaturation, and may act as surfactants and affect



the amphiphilicity of atmospheric particles (Su et al., 2022). Vehicle emissions might be their potential source. In contrast, biogenic VOC-derived (nitrooxy-)organosulfates have characteristics of short carbon chain ($C_{5-10}$), high degree of oxidation and double bond equivalent (DBE) values close to those of their biogenic precursors (Tao et al., 2014;Passananti et al., 2016). In addition, the H/C values of aromatic-like (nitrooxy-)organosulfates are relatively low, especially the polyaromatic

(nitrooxy-)organosulfates, which are considered to be mainly originated from anthropogenic emissions precursors (e.g., fireworks) (Xie et al., 2020a;Xie et al., 2020b;Kundu et al., 2013). Thus, both organosulfates and nitrooxy-organosulfates are prevalent in aerosol particles and have been identified as potential SOA markers and have substantial implications for atmospheric physicochemical processes (Kristensen and Glasius, 2011;Zhang et al., 2012;Zhu et al., 2019;Brüggemann et al., 2017;Froyd et al., 2010;Tolocka and Turpin, 2012;Surratt et al., 2008).

Tianjin is a typical coastal city in the North China Plain, where agriculture is developed and is susceptible to open biomass burning in autumn and winter, especially the agricultural fires (Fan et al., 2020). Under the effect of land and sea breezes, organic aerosols in Tianjin are greatly influenced by sea source, land source and diurnal chemistry (Fan et al., 2020). The low-molecular-weight organic compounds in aerosols, such as diacids, aliphatic lipids (*n*-alkanes, fatty acids, and fatty alcohols), and sugar compounds, have been investigated in Tianjin by gas chromatography-tandem mass spectrometry (Fan et al.,

2020;Pavuluri et al., 2020). However, there are limited studies on the molecular composition of high-molecular-weight organic compounds in ambient aerosols in Tianjin, especially the molecular markers with complex structure such as polycyclic aromatic hydrocarbons (PAHs) and polyacids emitted from biomass burning. In addition, little is known about organic sulfur-containing compounds, and the contribution of secondary transformation processes from biogenic and anthropogenic sources. Fourier transform ion cyclotron resonance mass spectrometry (FT-ICR MS) is known for its ultrahigh resolution and has been

applied to characterize natural organic mixtures in cloud water, rainwater, aerosols, and smoke particles emitted from biomass burning and coal combustion (Bianco et al., 2018;Mead et al., 2015;Song et al., 2018;Wu et al., 2019;Xie et al., 2020b;Qi et al., 2022;Han et al., 2022;Chen et al., 2022).

Therefore, the purpose of this study is to investigate the organic molecular composition with a wild range of molecular weight in urban aerosols using FT-ICR MS, and to better understand the impacts of secondary aerosol processes on the molecular

diversity in winter when biomass burning is generally active in the North Plain China.

## 2 Materials and Methods

### 2.1 Sample collection and dissolved organic isolation

The PM$_{2.5}$ sampling was carried out on the roof of a 20-meter-high building at Tianjin University in the Nankai District of urban Tianjin (39.11°N, 117.17°E). Other details about the samples had been described in a previous study (Fan et al., 2020).

Levoglucosan is treated as the primary tracer of biomass burning (Simoneit, 2002). Isoprene and pinene are selected as the predominant biogenic VOCs to indicate the photooxidation processes of plant and marine biogenic emission (Claeys et al.,





2007;Helmig et al., 2006;Sharkey et al., 2008). Based on our previous study, we have determined six isoprene oxides, including 2-methylglycolic acid, $C_5$-alkene triols and 2-methyltetrols, and four pinene oxides, such as 3-hydroxyglutaric acid (Fan et al., 2020). According to the sum of biogenic SOA tracers, we selected three groups of daytime and nighttime samples with strongly affected by biomass burning but with high, moderate, and low loadings of biogenic SOA (Table 1 and Figure 1) (i.e., High

SOA-loading - D: daytime sample with relatively high concentrations of SOA).

All the samples were analyzed by FT-ICR MS for the water-soluble organic compounds. The details of extraction and concentration of dissolved organic matters (DOM) from aerosol samples were taken from the previous study (Xie et al., 2020b). In short, the filter sample was sonicated in 10 mL ultrapure Milli-Q water for 10 min and repeated three times. The solution was then filtered with 0.45 μm hydrophilic PTFE filers. The extract was loaded onto a preconditioned solid-phase extraction

(SPE) cartridge (Oasis HLB, Waters, US). The cartridge was dried under a pure nitrogen flow. Then, the retained organics were eluted with HPLC-grade methanol.

**2.2 FT-ICR MS analysis**

The isolated organic compound fractions were analyzed with a solariX 2XR FT-ICR instrument (Bruker Daltonik GmbH, Bremen, Germany) equipped with a 7T superconducting magnet. Samples were ionized in negative ion mode using an

electrospray ionization (ESI) ion source. For full scan mass spectra, mass spectra were acquired from m/z 150 to 1000 (where most NOM measurements are conducted) with a transient size of 4 M words using the quadrupolar detection mode. A total of 256 individual transients were collected and co-added to enhanced signal-to-noise, resulting in resolving power of ~600,000 at m/z 400. The full scan mass spectra were internally calibrated using a series of homologous compounds in DataAnalysis (Bruker Daltonics). A peaks list with a signal-to-noise ratio (S/N) greater than four was generated. All possible formulae were

calculated using Composer 15.6 (Sierra Analytics) software with a mass tolerance of ± 0.5 ppm. A calculation criterion for the calculator was set as follows: $C_{50}H_{100}O_{50}N_2S_1$. All the calculated formulae with DBE greater than 30 were excluded.

The relative abundance weighted elemental ratios, $AI_{mod}$, $MW_w$, and $AI_{mod,w}$ were calculated based on previous studies (Zhao et al., 2013;Sleighter and Hatcher, 2008;Koch and Dittmar, 2016). All assigned molecular formulae were categorized into the following five classifications according to their elemental composition, (1) combustion-derived polycyclic aromatic

hydrocarbons (PAHs-like), (2) vascular plant-derived Polyphenols and PAHs with aliphatic chains (Polyphenols-like), (3) highly unsaturated and phenolic compounds (Phenols-like), (4) unsaturated aliphatic compounds (Aliphatics-like), and (5) carbohydrate, saturated fatty and sulfonic acids (Carbohydrates-like) (Merder et al., 2020;Šantl-Temkiv et al., 2013). According to the O/C ratio, the PAHs-like, Polyphenols-like, Phenols-like, and Aliphatics-like compound were derived into O-poor and O-rich classes (Table S1) (Merder et al., 2020).





# 3 Results and Discussion

## 3.1 General molecular characterization of organic aerosols

In this study, thousands of formulae (4995-6959) are obtained in each spectrum ranging from 150 to 1000 Da (Table S2). The identified molecular formulae are classified into CHO, CHON, CHOS, and CHONS components, based on their elemental compositions. For example, CHOS refers to formulae that contain C, H, O, and S elements. By comparing the number of four molecular components between all the samples, it was found that the S-containing species (CHOS and CHONS) are the most prominent components (61.8-96.5 %) in high and moderate SOA-loading samples, while only 25.2-37.6% of sample with low biogenic SOA-loading (Figure 2), suggesting that secondary transformation processes contributed significantly to S-containing compounds. Since there was little difference in the sum of biogenic SOA tracers between the nighttime samples in the moderate-high SOA loading groups, the molecular compositions in these two groups are similar overall (Figure2 and Figure3). The potential source of a compound class may be assessed by the ratio of the number- and intensity-weight of all compound classes, with a higher proportion indicating a greater contribution from the source. Aliphatics-like organics accounts for the highest proportion in high and moderate SOA-loading groups (43.8-50.5%), while Phenols-like and Aliphatics-like contribute the most in the low biogenic SOA-loading groups. In addition, combustion-derived and plant-derived aromatic compounds (PAHs-like and Polyphenols-like) in the low biogenic SOA-loading groups (34.5-38.8% and 30.5-33.7%) are significantly higher than the other two groups (Figure 3 and Figure S1). These indicate that intense biomass burning may contribute greatly to PAHs-like, Polyphenols-like and Phenols-like compounds. However, with the increase of biogenic SOA-loading, contributing more saturated Aliphatics-like and Carbohydrates-like compounds.

Table S2 summarizes the number of components in each subgroup and the relative abundance weighted elemental ratios, DBE, and $AI_{mod}$ for each sample (Zhao et al., 2013;Sleighter and Hatcher, 2008). $AI_{mod}$ index, based on heteroatoms such as oxygen, sulfur, and nitrogen, reflects C＝C double bond density to reveal the double-bond ratio to the total carbons in a molecule (Koch and Dittmar, 2006). As shown in Table S2, the $AI_{mod,w}$ values observes for different subgroups of each sample shown similar change trends: $AI_{mod,w}$ (CHON) > $AI_{mod,w}$ (CHO) > $AI_{mod,w}$ (CHOS) ≥ $AI_{mod,w}$ (CHONS). DBE values are also widely used to estimate the degree of unsaturation (Koch and Dittmar, 2006). In this study, the $DBE_w$ also have a similar pattern: $DBE_w$ (CHON) > $DBE_w$ (CHO) > $DBE_w$ (CHONS) > $DBE_w$ (CHOS). Trends in $DBE_w$ and $AI_{mod}$ are similar to previous studies, such as DOM from urban aerosols and biomass burning particulate matter (Song et al., 2018;Jiang et al., 2021). This might be because CHO and CHON compounds are mainly composed of combustion-derived highly unsaturated and phenolic compounds (Phenols-like), followed by highly aromatic PAHs-like and Polyphenols-like, while CHOS and CHONS were mostly Aliphatics-like, Phenols-like, and Carbohydrates-like with relatively high saturation (Table S3).


### 3.2 CHO compounds

CHO compounds, which may contain carboxyl and/or hydroxyl functional groups, have been widely detected in the ESI negative mode and identified in water-soluble organic mattes in aerosols and cloud water (Bianco et al., 2018;Tu et al., 2016;Xie et al., 2020a;Kourtchev et al., 2016). About 596 to 1967 ions could be assigned to CHO groups in the $PM_{2.5}$ samples,

accounting for 3.6% to 49.3% of the total compound in each sample, a difference of one order of magnitude (Figure 2, Table S2). Both the number and intensity-contribution of CHO compounds decrease significantly with increasing of biogenic SOA-loading, especially during the day, suggesting that biomass burning contributed greatly to the chemical diversity of CHO compounds.

As shown in Figure 4a, the CHO compounds are classified into 13 subgroups based on their O numbers. Most of the O > 10
subgroups are detected only in the samples with moderate to high biogenic SOA-loading, and the number increased with SOA concentrations. These high-oxygen-containing compounds are mainly highly unsaturated Phenols-like compounds (Figure 4b). Combined with DBE and carbon number plots (Figure S3), these unique O-rich compounds might be lignin-like compounds containing a single benzene ring, which are particularly sensitive to the UV light (Qi et al., 2016). In contrast, CHO compounds containing one oxygen atom existed only in the low SOA-loading group. Moreover, the number of each CHO subgroup at
night is much greater than that in day in moderate-high SOA-loading groups, and the opposite is true in the low SOA-loading group. These suggest that nocturnal chemistry was more efficient than photochemistry in oxidizing and forming biogenic secondary organic aerosols with high oxygen content at moderate-high SOA loadings, while photochemistry dominates the formation of secondary CHO compounds with low oxygen content at low SOA loadings.

$OS_C$ is an widely used parameter to describe the oxidation processes of complex organic mixtures (Kroll et al., 2011). The $OS_C$
values of semi-volatile and low-volatility oxidized organic aerosol (SV-OOA and LV-OOA) range from -1 to +1 and are less than 13 carbon atoms, which may be associated with multistep oxidation reactions. The $OS_C$ values of biomass burning organic aerosol (BBOA) is relatively low, ranging from -0.5 to -1.5, and greater than 7 carbon atoms. Molecules with $OS_C$ values less than -1 and carbon number greater than 20 may be related to hydrocarbon-like organic aerosol (HOA). As shown in Figure 5, the number of molecules in the SV-OOA and BBOA regions and their peak intensities increase significantly as the SOA-
loading increased, suggesting that the increase of SOA-loading might promote the multi-step oxidation reactions. Some of the high-intensity CHO compounds that in the SV-OOA area such as $C_{19}H_{28}O_7$, $C_{17}H_{26}O_8$, $C_{15}H_{18}O_8$, $C_{16}H_{24}O_8$, may be typical dimers of α-pinene secondary organic aerosol, as well as their homologues are detected in high and moderate SOA-loading samples. However, the relatively high-intensity CHO compounds such as $C_{20}H_{26}O_3$ and $C_{20}H_{30}O_2$, which had DBE values of 8 and 6, and may be diterpenoid derivatives (dehydroabietic acid and pimaric acid), are detected in the samples with low SOA-
loading (Gómez-González et al., 2012;Kourtchev et al., 2014;Yasmeen et al., 2011;Yasmeen et al., 2010;Kristensen et al., 2013;Kristensen et al., 2014;Müller et al., 2008). The most likely molecular structures of these α-pinene derivatives are illustrated in Figure 5. Obviously, the oxygen content and DBE values of these biogenic secondary CHO compounds in low SOA-loading groups are significantly lower than that of the other two moderate-high SOA-loading groups. These results





indicate that biogenic CHO compounds are mainly derivatives of monoterpenes, and the oxygen content of these biogenic SOA increases significantly with the increase of SOA-loading, especially monoterpene derivatives.

In addition, the effects of natural oxidation processes can also be observed at the microscale of individual peaks in the expanded segments of the full-scan mass spectra. For example, Figure 6 shows a ~ 0.3 Da segment from three daytime samples. The

5 unique CHO compounds are labelled between the three daytime samples, respectively. Two and one unique CHO compounds with higher oxygen-content were detected in the first and second spectra and transferred to species with lower mass (Figure 6a, b). On the contrary, there are five new CHO compounds with lower O-content in the spectrum of sample low SOA-loading - D, and the mass of these compounds increases with decreasing oxygen content (Figure 6c). The DBE value increased with the increase of oxygen number and $OS_C$, which is consistent with Figure 5. These results indicate that secondary CHO organic

aerosols have obvious bias in the formation processes. At low SOA load, especially monoterpene derivatives, secondary CHO organic aerosols are dominated by conjugated polyene compounds with low oxygen content and high saturation, while with the increased of SOA load, multi-step oxidation formed the O-rich compounds containing monophenyl ring, which may be important light-absorbing chromophores in the atmosphere (Deng et al., 2022).

### 3.3 CHON compounds

Large amounts of organic nitrogen compounds are observed in all $PM_{2.5}$ samples. CHON group could be assigned to 272 to 2513 ions in all samples, whose abundance-weighted contribution account for 1.0-25.3%, being similar to CHO (Table S2, Figure 2). Apparently, at the low SOA loading group, the abundance-weighted contribution of CHON compounds is significantly higher than that in the other two groups. Under the moderate-high SOA-loading, the concentration of levoglucosan, a marker of biomass burning, in the nighttime sample was about two times higher than that of the corresponding

daytime samples (Figure 1 and Table 1), indicating that the intensity of biomass burning was relatively high at night, the same as was the abundance-weighted contribution of CHON compounds. Combined with the fact that most CHON compounds are classified as O-poor phenols-like and polyphenols-like compounds (Figure 4d), which are similar to the characteristics of CHON compounds emitted from biomass materials (Song et al., 2018). Several of the highest intensity nitro-aromatic CHON compounds with C numbers less than 10, such as $C_7H_7N_1O_4$, $C_8H_9N_1O_3$, $C_8H_7N_1O_5$, $C_9H_9N_1O_5$, $C_7H_5N_1O_4$, which are detected

in particulate mattes emitted from combustion processes and potential contributors to light absorption as BrC chromophores (Song et al., 2018;Desyaterik et al., 2013;Yan et al., 2020;Iinuma et al., 2010) (Figure S4). The CHON compounds are classified into 24 subgroups based on their N and O numbers, including $N_1O_n$ ($N_1O_1$ - $N_1O_{13}$) and $N_2O_n$ ($N_2O_2$ - $N_2O_{12}$) subgroups (Figure 4c). 80%-100% of the CHON compounds have O/N ≥ 3. Hence, it can be inferred that most of CHON compounds in this study contained oxidized nitrogen functional groups such as nitro (-$NO_2$) and/or organonitrates (-$ONO_2$).

These results suggest that CHON compounds might be mainly derived from biomass burning, such as nitrophenols, nitrocatechols, nitroguaiacols, nitrosalycilic acids, which has also been observed in previous studies (Kourtchev et al., 2015;Zhang et al., 2013;Song et al., 2018).



The typical α-pinene and isoprene SOA components such as $C_{10}H_{14}N_1O_5$, $C_6H_{14}N_1O_7$, $C_5H_7N_1O_4$ (Perraud et al., 2010;Ng et al., 2008), are not detected in all samples, suggesting that isoprene and monoterpene may not contribute significantly to secondary CHON aerosols. Sesquiterpenes, however, might. Figure 7a shows the peak intensity distributions of seven nitrogen-containing SOA from β-caryophyllene (i.e., $C_{10}H_{13}N_1O_3$, $C_{12}H_{19}N_1O_6$, $C_{16}H_{27}N_1O_7$, $C_{15}H_{25}N_1O_8$, $C_{15}H_{27}N_1O_8$, $C_{17}H_{29}N_1O_8$,

$C_{15}H_{25}N_1O_9$) (Chan et al., 2010). The Moderate SOA-loading - D sample with highest concentrations of sesquiterpenes SOA have lowest relative abundance. In particular, the relative abundance of $C_{10}H_{13}N_1O_3$ in Low SOA-loading - N sample is about 3.5 time that of High SOA-loading - N sample. The relative abundance of these compounds in the nighttime samples is about twice that of the daytime samples. All these nitrogen-containing SOA compounds are detected in the series of β-caryophyllene/NO$_x$ irradiation experiments (Chan et al., 2010), but our study demonstrates that nocturnal chemistry might be

more conductive to sesquiterpene SOA compounds formation, especially under low sesquiterpene-loading conditions.

### 3.4 CHOS compounds

In our PM$_{2.5}$ samples, about 853 to 1663 ions are identified as CHOS, and the intensity contribution ranges from 20.4% to 78.8% (Figure 2, Table S2). Their intensity-contribution in the low SOA-loading samples (20.4-23.4%) is lower than those of the moderate-high biogenic SOA-loading groups (38.0-78.8%). The intensity-contribution of daytime samples is 23.9-25.4%

higher than that of corresponding nighttime samples at moderate ones (Figure 2a-d), while the opposite is true under the low SOA-loading, with a 3% higher intensity-contribution at night (Figure 2e, f). As shown in Table S2, the number of CHOS compounds is 853 in the Low SOA-loading - D sample, which nearly doubles with increase of SOA loads. The average of AI$_{mod,w}$ and DBE$_w$ values of CHOS compounds are significantly lower than that of CHO and CHON categories, and also much lower than that of CHOS generated by the combustion of coal (0.31) and biomass materials (0.13 - 0.18) (Song et al., 2018).

As shown in Figure 4e, the identified CHOS formulae are $O_3S$-$O_{14}S$ class species, with $O_{6-9}S$ being the most abundant. Interestingly, almost all the CHOS formulae had O/S ratios ≥ 4, and these CHOS compounds are tentatively regarded as organosulfates (OSs). The sulfate group ($OSO_3H$) carries four O atoms and readily deprotonates by ESI-, have been identified as contributing significantly to the generation of SOA (Wang et al., 2016;Lin et al., 2012;Tolocka and Turpin, 2012). Furthermore, the most abundant CHOS compounds such as $C_{15}H_{24}O_7S$, $C_{12}H_{24}O_5S$, $C_{15}H_{26}O_7S$, $C_{10}H_{18}O_6S$, $C_9H_{16}O_6S$,

$C_{10}H_{18}O_7S$, etc., and their corresponding homologs were detected (Figure 8 and Figure S5), which are generated by the oxidation of isoprene, monoterpene and sesquiterpene, respectively (Riva et al., 2016;Passananti et al., 2016;Surratt et al., 2008;Chan et al., 2010). These data indicate that the majority of CHOS compounds are derived primarily from biogenic VOCs oxidation, and that the formation efficiency of nocturnal chemistry and photochemistry varies with biogenic SOA-loading.
About half of CHOS compounds are Aliphatic-like compounds, followed by Phenols-like and Carbohydrates-like compounds

with low aromatic degree (Figure 4f). Not only the total formulae number, but also the number of each CHOS subgroup in the moderate-high SOA-loading daytime samples are significantly higher than that in the corresponding nighttime samples. In contrast, in the case of low biogenic SOA loading, it is opposite. Interestingly, the O ≥ 12 organosulfates are identified only in



the moderate-high biogenic SOA-loading groups, and the formula number is even slightly higher in the moderate SOA-loading group than that in high biogenic SOA-loading group (Figure 4e). And 305 to 560 OSs compounds with high O/S ratios ($\geq$ 10) are found to be densely distributed in the moderate-high biogenic SOA-loading groups compared to the samples with low SOA loads, particularly in the region of high-molecular-weight (HMW > 500 Da) (Figure 8). Similarly, the number of these HMW OSs is 1.6-2.3 times higher in the daytime samples with moderate-high biogenic SOA loads than in the nighttime samples, but only 33% in the low SOA loads. Apparently, these high-oxygen-containing compounds are mainly composed of O-rich Phenols-like and Aliphatics-like compounds (Figure 4f), unlike long-chain aliphatic organosulfates with a few or no additional functional groups, which are emitted from traffic (Tao et al., 2014). Combined with DBE value and C number (Figure 8 and Figure S5), it could be inferred that these O-rich species might have alicyclic alkane organosulfates containing conjugated polyene, similar to organosulfates derived from biogenic VOCs oxidation (Chan et al., 2010;Surratt et al., 2008). The differences between daytime and nighttime aerosols indicate that photochemical oxidation should be more beneficial for multi-step oxidation of biogenic organosulfates when the biogenic SOA loading is relatively high. On the other hand, organosulfates are more likely to be generated by nocturnal chemistry when the biogenic SOA loading is low.

In addition, it should be noted that $C_5$ organosulfates ($C_5H_{10}O_5S$, $C_5H_{10}O_6S$ and $C_5H_{12}O_7S$, etc.) typically associated with isoprene ($C_5H_8$) in laboratory studies (Surratt et al., 2008;Chan et al., 2010), are not observed in all samples. Although the corresponding $C_{6-9}$ isoprene-related organosulfates homologs are detected, their relative abundance is low (Figure 8 and Figure S5). In addition, the concentrations of isoprene and sesquiterpene SOA tracers were similar in the moderate-high SOA loading groups, but monoterpene SOA differed greatly. These results suggest that isoprene may be a relatively minor contributor to the population of organosulfates in winter, and monoterpene might be the potential contributors to the high-oxygen-containing OSs.

### 3.5 CHONS compounds

The intensity of CHONS compounds accounts for 5.2-24.3% of total compounds in all $PM_{2.5}$ samples. Both the intensity weighted and number increased with the increasing of biogenic SOA loads (Figure 2, Table S2). The average OM/OC ratios of CHONS are much higher than other subgroups (Table S2), which is consistent with S atoms in molecules, indicating that the oxidation time of CHONS compounds is longer or the oxidation efficiency is higher (Altieri et al., 2009). As the increased of biogenic SOA loading, the number of CHONS compounds increases dramatically by 1294, implying that biogenic SOA contributes significantly to the chemical diversity of CHONS compounds. Similar to CHOS compounds, the total number of CHONS compounds in the moderate-high SOA-loading daytime samples is 436 more than in the nighttime samples, whereas in the low SOA loads group, the number of CHONS compounds in the daytime sample is 55% that in the nighttime sample.

Based on the N and O atoms, CHONS compounds were classified into 22 subgroups, including $N_1O_nS_1$ ($N_1O_3S_1$- $N_1O_{14}S_1$) and $N_2O_nS_1$ ($N_2O_5S_1$- $N_2O_{14}S_1$) (Figure 4g). It should be noted that more than 70% $N_1O_nS_1$ formulae contain seven or more O atoms, and about 50% $N_2O_nS_1$ formulae have fewer than ten O atoms, implying that these CHONS compounds are probably





nitrooxy-organsulfates (nitrooxy-OSs) containing nitrate (-ONO$_2$) groups (Figure 4g). CHONS compounds are mainly Phenols-like and Aliphatics-like, followed by Carbohydrates-like (Figure 4h), suggesting that these compounds might contain long alkyl carbon chains character. Similar to CHOS compounds, CHONS compounds might be formed primarily by the secondary conversion processes of VOCs at high concentrations of nitrogen oxide (NO$_x$) (Surratt et al., 2008;Kundu et al.,

2013). Not only the total formulae number, but also the number of each CHONS subgroup of daytime samples are higher than that of corresponding nighttime samples at moderate and high SOA load, while the number is opposite at low SOA load (Figure 4g, h), suggesting that nocturnal chemistry is more conductive to nitrooxy-OSs generation at low biogenic SOA loads, while photochemistry is more efficient for nitrooxy-OSs formation with the increasing of biogenic SOA loads.

Figure 9 shows the DBE, C, and O atomic distributions in the CHONS compounds. The most abundant nitrooxy-OSs

$C_{10}H_{17}N_1O_7S_1$, $C_{10}H_{17}N_1O_9S_1$, $C_{10}H_{19}N_1O_9S_1$, $C_{15}H_{25}N_1O_7S_1$, $C_{10}H_{18}N_2O_{11}S_1$, which are generated by oxidation of α-Terpinene, α, β-Pinene, β-Caryophyllene, and Terpinolene in atmosphere and smog-chamber experiments (Altieri et al., 2008;Kundu et al., 2013;Lin et al., 2012;Surratt et al., 2008;Chan et al., 2010;Wang et al., 2021), as well as the corresponding homologs, are detected in all samples, highlighting the importance of biogenic VOCs to form CHONS compounds. It worth noting that in the two low SOA-loading samples, the highest abundance nitrooxy-OSs with *m/z* 294.0653 and molecular formula

$C_{10}H_{17}N_1O_7S_1$ might be formed by oxidation of α-pinene in the presence of SO$_2$ and NO$_x$ (Kundu et al., 2013;Altieri et al., 2008), suggesting that CHONS compounds might be mainly derived from the oxidation of monoterpene when SOA loading is low, especially pinonic acid and pinic acid.

Interestingly, most of the $N_1O_{12-14}S_1$ and $N_2O_nS_1$ species were detected only in the moderate and high SOA-loading samples, and the number of these compounds in daytime samples was more abundant than in the nighttime samples (Figure 4g).

According to Table 1, the sum concentrations of isoprene SOA tracers in the moderate and high SOA-loading groups are similar, but the concentrations of monoterpene SOA tracers were much different. Most of the relative-high-abundance CHONS compounds are monoterpene nitrooxy-OSs. Therefore, our results indicate that the load of biogenic SOA is an important factor determining the oxidation degree, nitrate (-ONO$_2$) content and chemical diversity of CHONS compounds.

In particular, previous studies have shown that nitrate (-ONO$_2$) and/or sulfate (-OSO$_3$H) might undergo hydrolysis in the

presence of atmospheric water (Lin et al., 2012;Liu et al., 2012;Hu et al., 2011). Figure 7b shows one of the hydrolysis reactions of nitrooxy-OSs with relative high abundance. Obviously, the terpene-related nitrooxy-OSs ($C_{10}H_{18}N_2O_{11}S_1$) were only observed in the moderate and high SOA-loading groups. In the presence of water, the terpene-related nitrooxy-OSs substitutes the nitrate group with hydroxyl groups by hydrolysis. Therefore, it can be inferred that the corresponding CHOS and CHONS (N = 1) organosulfates in the studied samples may be generated though the hydrolysis of nitrate of nitrooxy-OSs.





# 4 Conclusions

Four categories of organic compounds, including CHO, CHON, CHOS, and CHONS species in the urban Tianjin in winter were determined by ultrahigh-resolution FT-ICR MS. Biomass burning was found to contribute significantly to CHO and CHON compound. Most of the CHO and CHON compounds are O-poor and highly unsaturated PAHs and (Poly)phenols, which are important light-absorbing chromophores in the atmosphere. There is a significant change for both the number and abundance-weight contribution between daytime and nighttime samples at different biogenic SOA loadings. The nocturnal chemistry is more efficient than photochemistry in oxidizing and forming secondary CHO and CHON compounds with high oxygen content at moderate-high SOA loadings, while photochemistry dominates the formation of secondary CHO and CHON compounds with low oxygen content at low SOA loadings. The biogenic CHO compounds are mainly derivatives of monoterpenes. However, nocturnal chemistry might be more conductive to sesquiterpene SOA formation, especially under low sesquiterpene-loading conditions.

The S-containing compounds (CHOS and CHONS) are mainly derived from oxidation of biogenic VOCs. About 96% of S-containing compounds are considered as organosulfates and nitrooxy-organosulfates. Compared with CHO and CHON compounds, high abundances of S-containing compounds with higher H/C ratio and lower DBE and $AI_{mod}$ values are mainly composed of by alicyclic alkane organosulfates. With the increasing SOA loading, the contribution of the number and abundance-weight contribution of S-containing compounds increased dramatically. The nocturnal chemistry is more conductive to nitrooxy-OSs generation at low biogenic SOA loadings, while photochemistry is more efficient for nitrooxy-OSs formation with the increasing of biogenic SOA. Monoterpenes might be potential contributors to high-oxygen-content organosulfates. Our results show that the biogenic SOA is an important factor determining the oxidation degree, nitrate (-$ONO_2$) content and chemical diversity of S-containing compounds in urban Tianjin. Moreover, some of the CHOS and CHONS (N = 1) organosulfates can also be formed by the hydrolysis of nitrate group of nitrooxy-organosulfates.

*Data availability.* The data is available upon request from the corresponding author.

*Author contribution.* PF designed the study. SZ, SC, JD, YF, and QX carried out the experiments and performed the data analysis. SZ prepared the first version of the manuscript with contributions from all co-authors. All authors verified the final version of the manuscript.

*Competing interests.* The authors declare that they have no conflict of interest.

*Acknowledgements.* The authors greatly appreciate the assistance of editor and three anonymous reviewers for the helpful comments that greatly improved the quality of this manuscript.

*Financial support.* This research was supported by the National Natural Science Foundation of China (Grant Nos. 42130513 and 41905110) and the Strategic Priority Research Program of the Chinese Academy of Sciences (No. XDA23020301).



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



Table 1. The concentrations (ng·m$^{-3}$) of chemical compounds in PM$_{2.5}$ samples.

| Organic marker compounds | | | High SOA-loading | | Moderate SOA-loading | | Low SOA-loading | |
|---|---|---|---|---|---|---|---|---|
| | | | D[a] | N | D | N | D | N |
| Sum of biogenic SOA tracers | | | 68.7 | 28.4 | 48.9 | 31.2 | 15.9 | 7.13 |
| Biogenic SOA traces | Isoprene SOA traces | 2-methyglyceric acid | 2.28 | 1.12 | 2.35 | 0.88 | 1.58 | 0.03 |
| | | C$_5$-alkene triols | 1.25 | 0.24 | 2.48 | 0.99 | 0.24 | 0.79 |
| | | 2-methylthreitol | 1.71 | 0.91 | 1.51 | 3.46 | 0.46 | 0.20 |
| | | 2-methylerythitol | 4.12 | 1.21 | 1.67 | 3.72 | 0.46 | 0.48 |
| | | Subtotal | 9.37 | 3.47 | 8.02 | 9.04 | 2.73 | 1.50 |
| | Monoterpene SOA tracers | 3-hydroxyglutaric acid | 1.76 | 1.46 | 2.27 | 1.16 | 0.53 | 0.74 |
| | | Pinonic acid | 15.4 | 9.57 | 9.53 | 5.76 | 5.29 | 0.55 |
| | | Pinic acid | 24.9 | 7.71 | 13.8 | 6.86 | 0.89 | 0.58 |
| | | MBTCA[b] | 0.20 | 0.19 | 1.24 | 2.05 | 0.12 | 0.12 |
| | | Subtotal | 42.3 | 18.9 | 26.8 | 15.8 | 6.71 | 1.13 |
| | Sesquiterpene SOA tracer | β-caryophyllinic acid | 17.0 | 6.05 | 14.1 | 6.37 | 6.50 | 6.00 |
| Biomass burning tracers | Levoglucosan | | 471 | 258 | 167 | 303 | 127 | 424 |
| | Galactosan | | 49.9 | 22.4 | 31.3 | 16.2 | 13.4 | 44.4 |
| | Mannosan | | 54.1 | 25.8 | 46.3 | 25.6 | 22.5 | 69.3 |
| | L/M ratio | | 8.71 | 10.0 | 3.61 | 11.8 | 5.64 | 6.12 |

Note: [a]D: daytime, N: nighttime; [b]MBTCA: 3-methyl-1,2,3-butanetricarboxylic acid.



**Figure 1. Temporal variations in the concentrations of biogenic SOA tracers and biomass burning tracer detected in Tianjin PM₂.₅ (Fan et al., 2020). (a) Sum of biogenic SOA tracers; (b) Isoprene SOA tracer; (c) Monoterpene SOA tracers; (d) β-Caryophyllene SOA tracer. The red, green, and blue circles represent the PM₂.₅ samples with high, moderate, and low SOA-loading, respectively.**





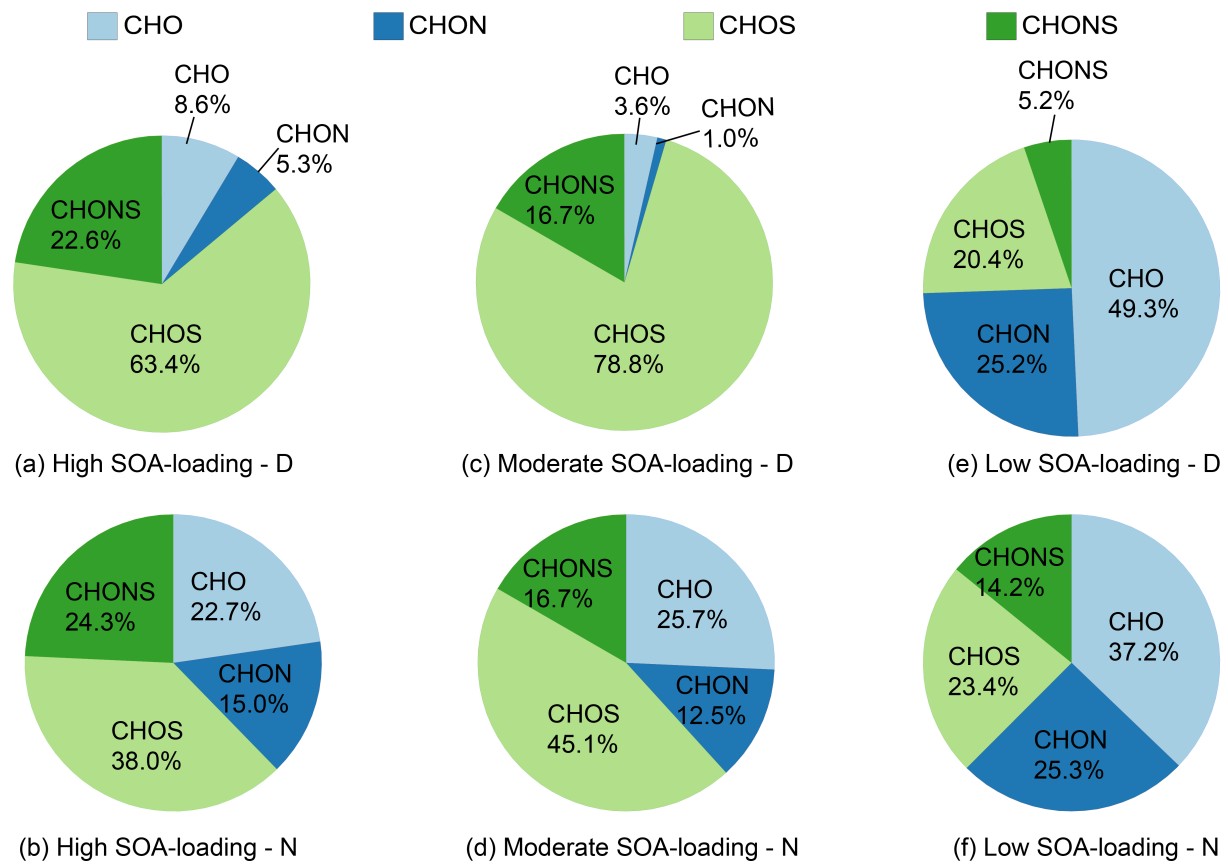

**Figure 2. Comparison of molecular elemental types of all PM$_{2.5}$ samples. The pie chart shows the percentage of the different compound groups in various samples by intensity.**





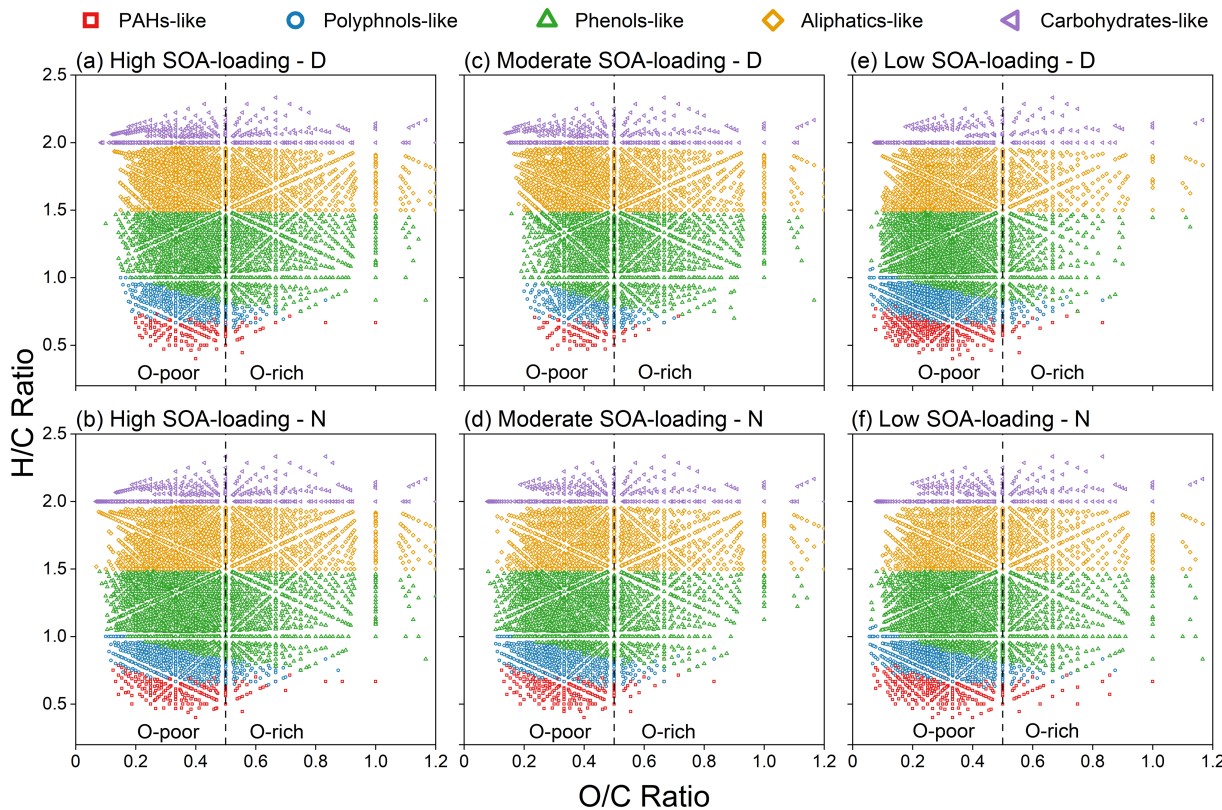

**Figure 3. The van Krevelen diagrams of five compounds classes in each sample. (1) PAHs-like: polycyclic aromatic hydrocarbons, (2) Polyphenols-like: polyphenols and PAHs with aliphatic chains, (3) Phenols-like: highly unsaturated and phenolic compounds, (4) Aliphatics-like: unsaturated aliphatic compounds, (5) Carbohydrates-like: carbohydrate, saturated fatty and sulfonic acids. Note: oxygen-poor compounds (O/C ≤ 0.5), oxygen-rich compounds (O/C > 0.5).**







**Figure 4.** (a, c, e, f) Classification of CHO, CHON, CHOS, and CHONS compounds into different subgroups based on the number of O and N atoms in molecules. (b, d, f, h) The number of five compound classifications of all molecules to each sample. The column is the sum of formula number in each subgroup. The 'n' is the number of oxygen atoms, respectively.





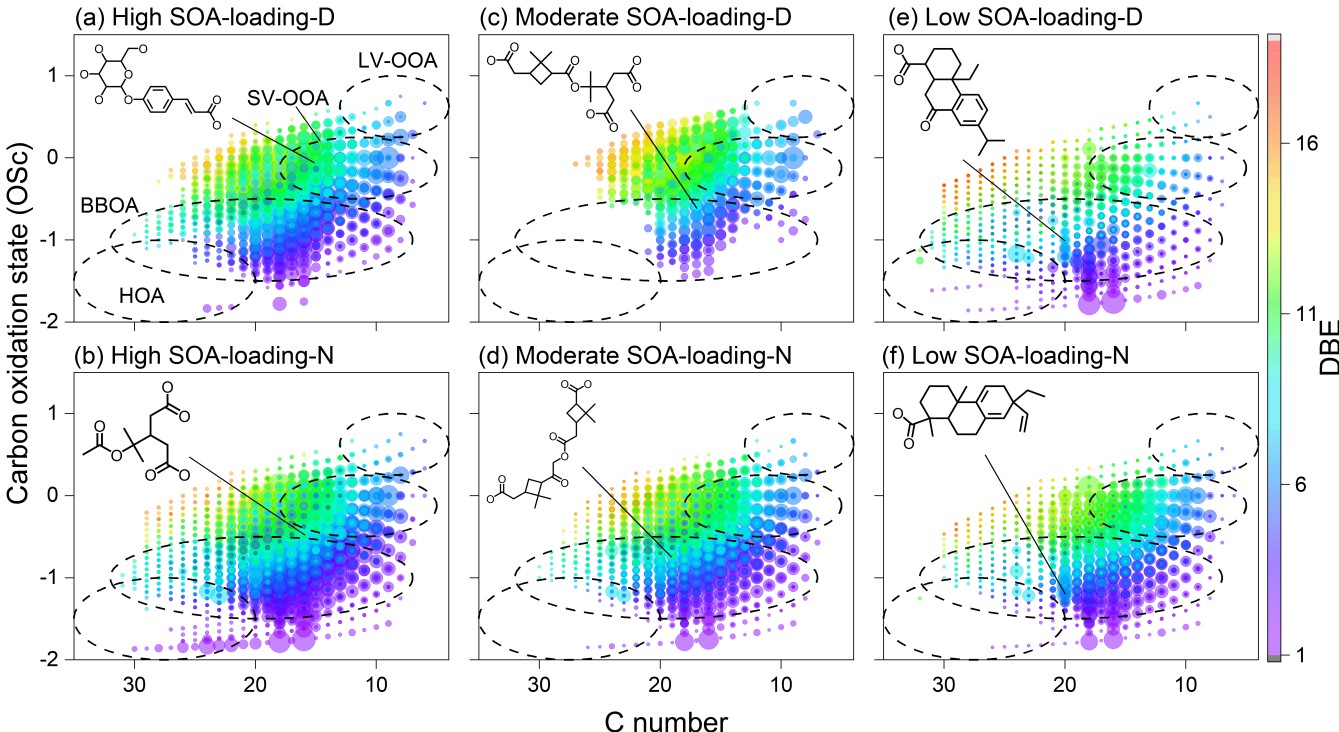

**Figure 5. Carbon oxidation state (OS_C) versus C number of CHO compounds. The size and colour bar denote the relative peak intensity and DBE value. The dash circles are marked as SV-OOA (semi-volatile oxidized organic aerosol), LV-OOA (low-volatility oxidized organic aerosol), BBOOA (biomass burning organic aerosol), and HOA (hydrocarbon-like organic aerosol). The formulae for biogenic SOA compounds with relative high intensity were $C_{15}H_{18}O_8$, $C_{16}H_{24}O_8$, $C_{19}H_{28}O_7$, $C_{17}H_{26}O_8$, $C_{20}H_{26}O_3$, $C_{20}H_{30}O_2$, respectively. Note that the proposed structures were representative, not determined.**



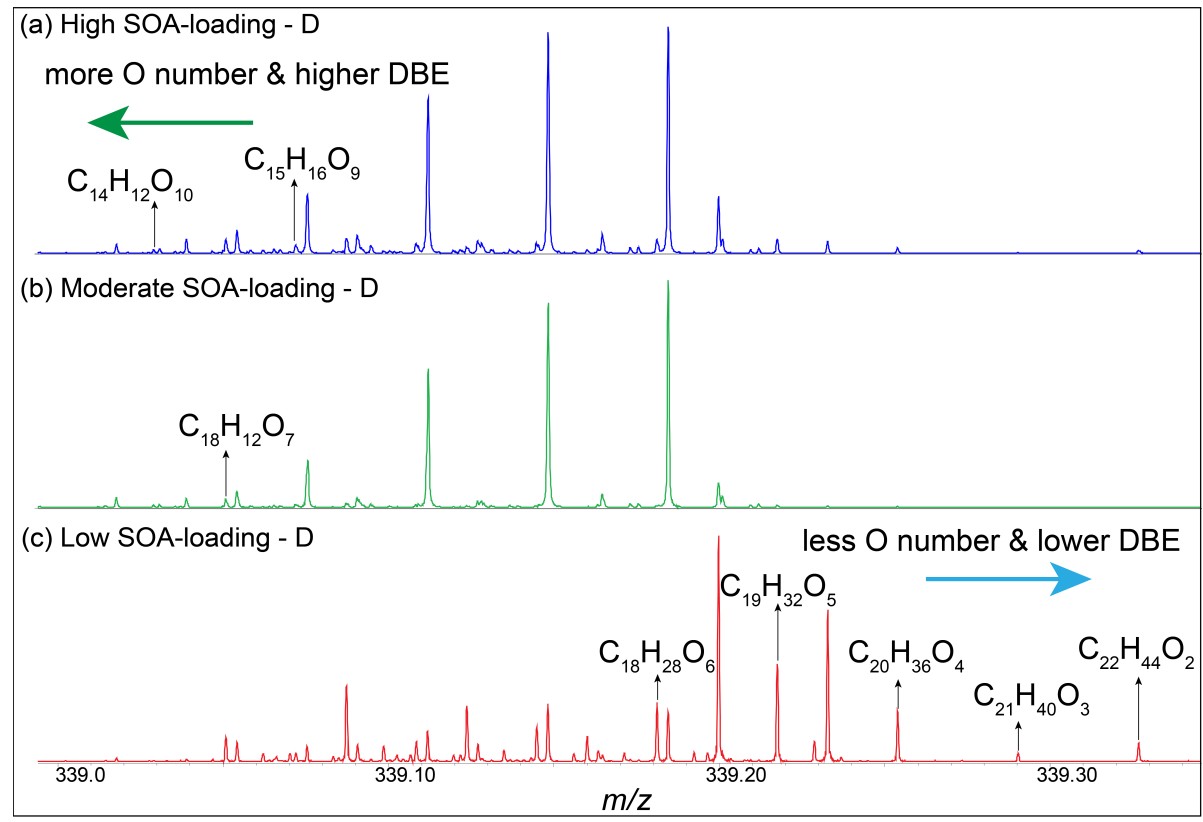

**Figure 6. Mass-scale-expanded segments (0.30 Da) of the broadband mass spectra for CHO compounds. The shift to higher mass defect in the low SOA-loading - D sample reflects lower oxygen content and DBE value, and vice versa reflects higher oxygen content and DBE value in high and moderate SOA-loading - D sample.**



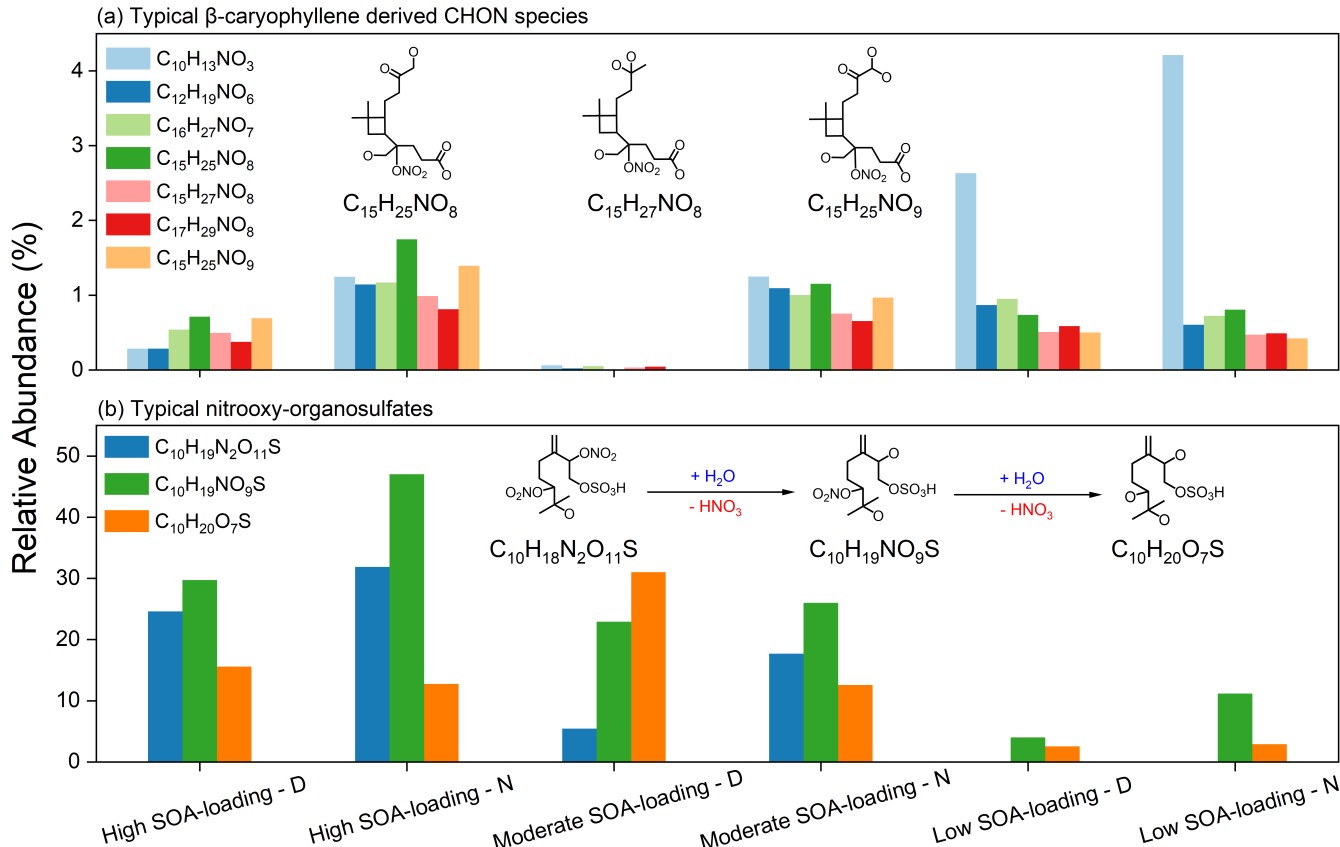

**Figure 7. (a) Relative abundance distributions of typical β-caryophyllene derived CHON compounds. Some of the proposed chemical structures have been reported in previous study (Chan et al., 2010). (b) Relative abundance distributions of typical nitrooxy-organosulfates. The hydrolysis reactions and the proposed chemical structures have been reported in previous study (Lin et al., 2012).**

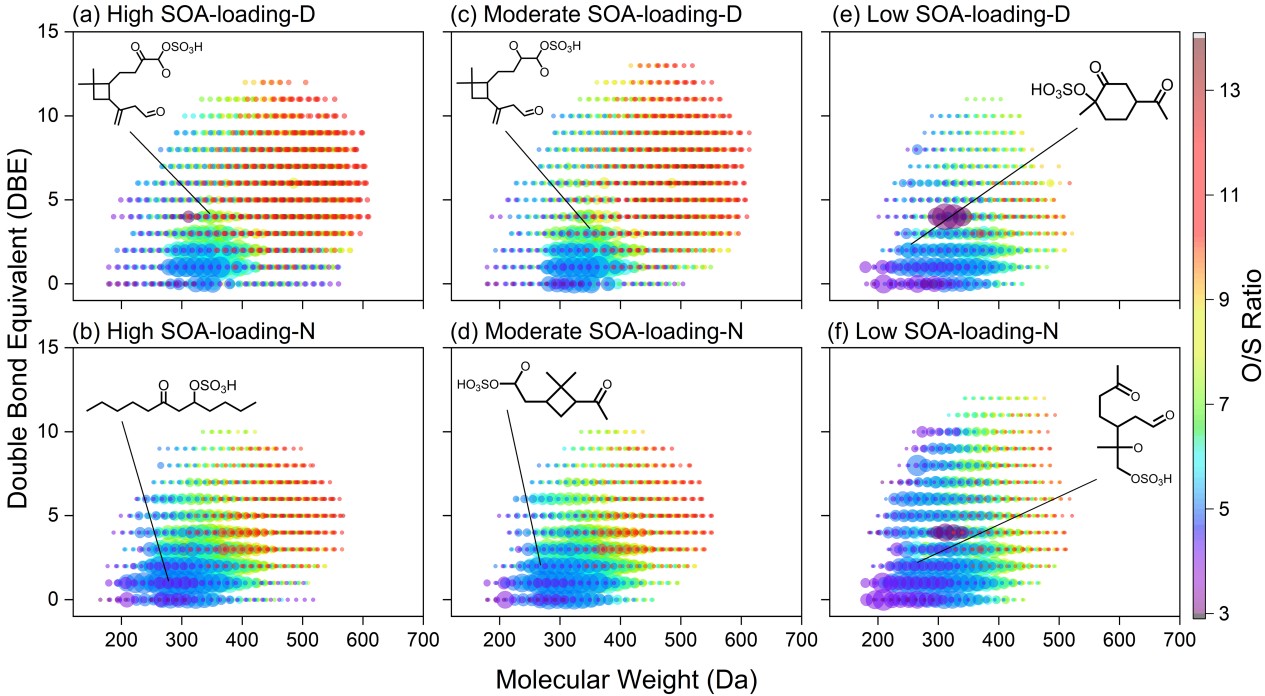

**Figure 8. DBE vs. molecular weight of CHOS compounds. The colour bar and marker size denote the O/S ratios and the relative peak magnitudes of CHOS compounds. The formulae for biogenic SOA compounds with relative high intensity were $C_{15}H_{24}O_7S$, $C_{12}H_{24}O_5S$, $C_{15}H_{26}O_7S$, $C_{10}H_{18}O_6S$, $C_9H_{16}O_6S$, $C_{10}H_{18}O_7S$. Note that the proposed structures were representative, not determined.**


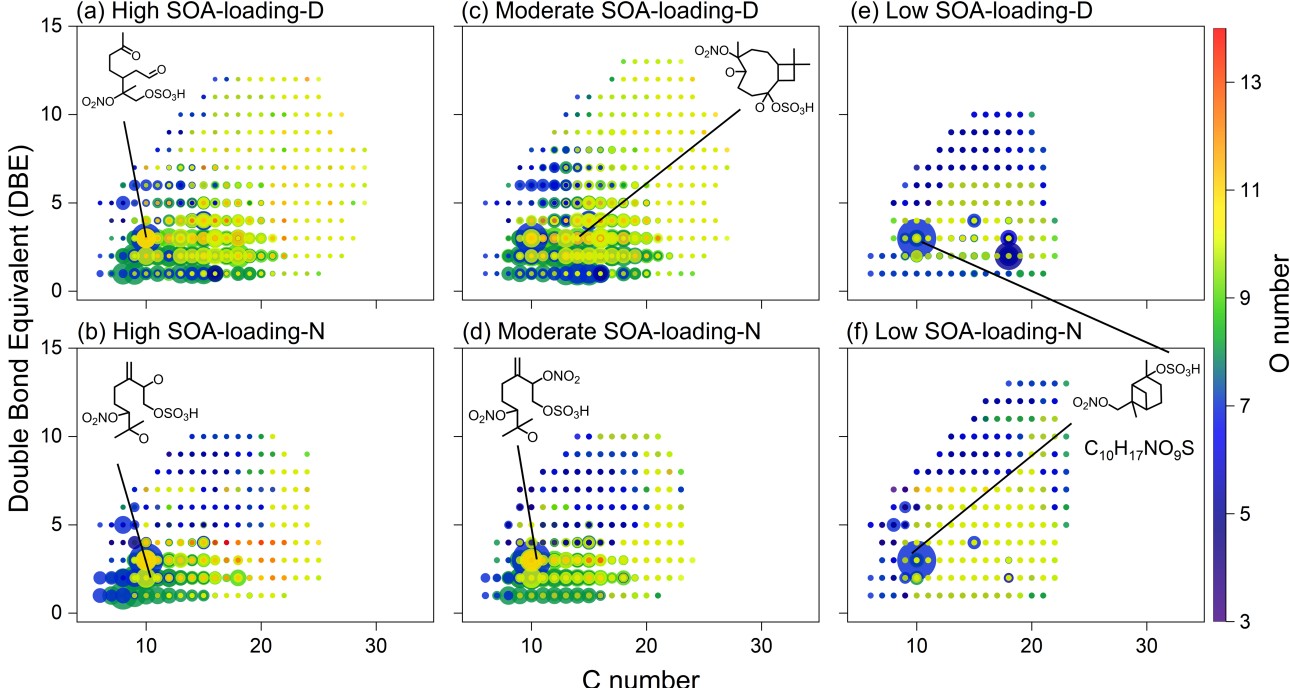

**Figure 9. Double bond equivalent (DBE) vs. C number for all the CHONS compounds. The colour bar and marker size denote the number of O atoms and the relative peak intensities of molecular formulae on a logarithmic scale. Note: the proposed structure is representative, not determined.**