# Peer review of "Impact of biogenic SOA loading on the molecular composition of wintertime PM2.5 in urban Tianjin: an insight from Fourier transform ion cyclotron resonance mass spectrometry"

_Atmospheric Chemistry and Physics, 2022_

## Referee Comment (RC2)

[referee-annotated manuscript omitted]

---

## Author Comment (AC1)

**Responses to Reviewer #1**

General comments.

This paper investigated the molecular compositions of urban organic aerosols by ultra-high resolution FTICR MS. The results give an insight into the sources of organic aerosols with different SOA loadings in urban Tianjin, China on a molecular level. The data of this kind is informative and valuable and related discussions would inspire some new aspects for further OA studies. While some statements and conclusions in this study need to be described more clearly. After addressing the following issues, it is suitable to be published on ACP.

Re: We appreciate the valuable comments from the reviewer. We have made changes/modifications to both the main text and the supplemental information. Detailed responses are shown below.

**Comments**

1. My major concern is the sample selection and classification. Discussion and results in this work are too much dependent on the sample selection and classification. It is possible to draw a opposite conclusion with different samples or other classification way. A simple statement of "according to biogenic SOA tracers…." in section 2.1 is not very sufficient.

For example, the high SOA-loading sample is not the one with highest SOA tracer concentration, especially for the nighttime one (Figure 1). The daytime high SOA-loading sample also accompanied with highest levoglucosan concentration, how to evaluate the variation of relative contribution from biomass burning and secondary formation? Is it better to make the classification based on the tracer contribution among total organic aerosol concentration? What's more, the biogenic SOA concentrations in the night moderate SOA-loading sample are comparable or slightly higher than the high SOA-loading one. I may suggest a more detailed statement on the sample selection.

Re: Thanks. Actually, we distinguish different SOA loading based on the subtotal concentrations of biogenic SOA tracers of daytime aerosols, and then select the corresponding nocturnal one. And a more detailed description of samples selection is also supplemented and modified (lines 4-7, page 4).

"According to the sum of biogenic SOA tracers of daytime aerosols, we selected three groups with strongly affected by biomass burning but with high, moderate, and low loadings of biogenic SOA, then selected the corresponding nocturnal ones (Table 1, Figure 1) (i.e., High SOA-loading - D: daytime sample with relatively high concentrations of SOA)."

According to previous studies, it is known that the North China Plain (NCP) is considered to be one of the areas with the highest amount of biomass burning and highest anthropogenic emissions in the world. Tianjin is the largest coastal city in the NCP (Fan et al., 2020;Sun et al., 2013;Zhu et al., 2016). Thus, it means the aerosol samples collected in winter are normally influence by biomass burning. The impact of biogenic SOA loadings on molecular composition is investigated with high biomass burning contribution in this study. The products of biomass burning are mainly CHO and CHON compounds(Song et al., 2018). Thus, the contribution of biomass burning is evaluated by the ratios of these two.

2 In the section of Materials and Methods, some important information directly related to the discussion needs to be added.

(1) The discussion on compound categorization throughout the main text, which support some key points in the work. Only a very short statement (lines 24 in page 4) on compound categorization referring another two papers may not a good way.

Re: Thanks for the good suggestion. Some critical parameter information have been added (lines 27-30, page 4). Moreover, detailed classification criteria for compounds can be found in Table S1 of the supplementary material.

"(1) combustion-derived polycyclic aromatic hydrocarbons (PAHs-like; AImod > 0.66), (2) vascular plant-derived Polyphenols and PAHs with aliphatic chains (Polyphenols-like; 0.50 <

AImod ≤ 0.66), (3) highly unsaturated and phenolic compounds (Phenols-like; AImod ≤ 0.50 and H/C < 1.5), (4) unsaturated aliphatic compounds (Aliphatics-like; 1.5 ≤ H/C < 2), (5) carbohydrate, saturated fatty and sulfonic acids (Carbohydrates-like; H/C ≥ 2)"

(2) Ion suppression in the ionization source of FTICR MS could influence the ionization efficiency a lot. How much would this effect the further discussion in this work?

Re: Referring to previous studies, normalize all mass spectral peaks with the highest peak in each mass spectrum to reduce the effect of ion suppression in the ionization source (Song et al., 2018;Xie et al., 2020).

(3) Sampling strategy needs to be briefly introduced in this section. "Described in a previous study" (line 29 in page 3) may be not a clear statement.

Re: Thanks for the good suggestion. The sampling strategy has been simplified (lines 26-28, page 3; lines 1-2, page 4).

"In this study, we selected wintertime PM2.5 samples collected in urban Tianjin, East China during November – December 2016, as a part of the wintertime campaign of the Atmospheric Pollution and Human Health in a Chinese Megacity (APHH-Beijing) programme"

"Details of these PM2.5 samples as well as molecular compositions of organic aerosols were provided in our previous study"

3 Some conclusion in section 3.3 and 3.4 may need more evidence.

(1) In section 3.3, authors draw a conclusion that biomass burning as the major source of CHON compounds. Why the sample with highest levoglucosan concentration (high SOA loading day sample) show very low CHON contribution (figure 2a), while low-levo sample (low SOA loading day sample) with high CHON contribution (figure 2e).

Re: The relative abundance of mass spectral peak is strongly dependent on the ionization capability and not only related to concentration, especially in complex matrix with sequential acquisition. Organosulfates are more ionized than CHO and CHON

compounds (Bianco et al., 2018). In addition, the organosulfates are mainly generated by the oxidation of biogenic VOCs, and its quantity increases with the increase of SOA loads. Thus, the sample with highest levoglucosan concentration show very low CHON contribution (figure 2a), while low SOA loading day sample with high CHON contribution (figure 2e).

(2) In section 3.4, authors suggest CHOS from biogenic VOC oxidation (line 27 page 8), while biogenic VOC emissions should be very low in winter.

Re: Not only the intensity contribution of CHOS compounds, but also the number of CHOS compounds increase with the increase of biogenic SOA tracers loads. In addition, previous studies have found that the oxidation of biogenic VOCs is an important cause of organosulfates (Surratt et al., 2008;Altieri et al., 2009). On the other hand, CHOS compounds with the largest intensities and number of homologues (Figure S6, page 12, supplemental information) are also formed by the oxidation of biogenic VOCs. Therefore, even if the emission of biogenic VOCs is low in winter, the contribution to organosulfates is still high.

[Figure]

Figure S6. Two-dimensional Kendrick mass defect (KMD) matrix plot for CHO, CHON, CHOS and CHONS compounds of High SOA-loading-D sample. The KMD [O] denotes the Kendrick mass defect of hydroxyl functional group (-OH). The KMD [CH$_2$] denotes the Kendrick mass defect of methylene group (CH$_2$). The magnified star symbols represent C$_{16}$H$_{24}$O$_8$, C$_{15}$H$_{25}$NO$_8$, C$_{15}$H$_{24}$O$_7$S and C$_{10}$H$_{17}$NO$_9$S with relative high abundance, respectively.

(3) The statements in lines 18 page 8 and lines 10-13 page 9 need more evidence. Could the author show other evidence or data to support this conclusion?

Re: Thanks for the good suggestion. We have added the KMD plots (Figure S6) (page 12, supplemental information). For example, in the horizontal direction, different number of hydroxyl functional groups (-OH) are added to organic compounds, which results in multi-step oxidation.

**Minor comments**

1 Figure 1: I think the data in Figure 1 is from a published paper (Fan et al., 2020). It should be better to show Figure 1 in supplementary.

Re: This figure can indeed be removed from the supplementary material, but placing it in the main text allows the reader to see the characteristics of the sample at a glance and increases the readability of the paper.

2 Page 1 line 20-23: this sentence needs to be re-organized. How to arrive the conclusion of "sensitive to ...chromophores"?

Re: Thanks for the good suggestion. We have modified this sentence (lines 20-22, page 1).

"Our results show that most of the CHO and CHON compounds are derived from biomass burning, which are O-poor and contain aromatic rings that probably contribute to light-absorbing brown carbon (BrC) chromophores."

3 Page 1 Line 25: Why photochemistry processes produce compounds with low oxygen content under low SOA loading condition? SOA should generally be more oxidized.

Re: For one thing, the FT-ICR MS used in this study has ultra-high resolution, can detect some trace organic compounds. This is something that could not be reached with GC-MS. For another, these low-oxygen-content compounds form homologues primarily through methylene ($CH_2$), and the longer the carbon chain, the lower the oxidation (Figure S6).

3 Page 1 lines 30-31: this sentence is a little confusing. Suggest to be clearer.

Re: Thanks for the good suggestion. We have modified this sentence (lines 29-30, page 1).

"Terpenes may be potential major contributors to organosulfates and nitrooxy-organosulfates."

4 Page 5 line 5: change "of" to "for"

Re: Thanks for the good suggestion. But we don't think this word is wrong.

5 Page 5 lines 12-16: These sentences are unclear.

Re: Thanks for the good suggestion. We have modified this sentence (lines 12-14, page 5).

"Aliphatics-like organics account for the highest proportion in high and moderate SOA-loading groups (43.8-50.5%). In contrast, Phenols-like and Aliphatics-like contribute the most in the low biogenic SOA-loading groups (68.4-69.3%)."

6 Page 6 Lines 4-5: Do you mean the compound number or contribution with a difference of one order of magnitude? Pls be clear.

Re: Thanks. It's actually an order of magnitude difference in intensity contribution. We have modified this sentence (lines 4-6, page 6).

"About 596 to 1967 ions could be assigned to CHO groups in the PM2.5 samples (Table S2). The intensity contribution of CHO compounds accounted for 3.6% to 49.3% of the total compound in each sample, a difference of one order of magnitude (Figure 2)."

7 Page 6 Lines 6-7: Based on figure 2, the contribution of CHO compounds increased from moderate SOA loading sample to high SOA loading sample.

Re: Thank you. According to previous studies, the L/M ratio can used to indicate different sources of biomass burning (Table 1) (Engling et al., 2014;Engling et al., 2009). The CHO compounds emitted by different biomass materials burning are different (Song et al., 2018). Thus, although the concentration of levoglucosan in High SOA-loading-D sample was higher than Moderate SOA-loading-D, the intensity contribution of CHO compounds was slightly different.

References:

Altieri, K. E., Turpin, B. J., and Seitzinger, S. P.: Oligomers, organosulfates, and nitrooxy organosulfates in rainwater identified by ultra-high resolution electrospray ionization FT-ICR mass spectrometry, Atmospheric Chemistry and Physics, 9, 2533-2542, 10.5194/acp-9-2533-2009, 2009.

Bianco, A., Deguillaume, L., Vaitilingom, M., Nicol, E., Baray, J. L., Chaumerliac, N., and Bridoux, M.: Molecular Characterization of Cloud Water Samples Collected at the Puy de Dome (France) by Fourier Transform Ion Cyclotron Resonance Mass Spectrometry, Environ Sci Technol, 52, 10275-10285, 10.1021/acs.est.8b01964, 2018.

Engling, G., Lee, J. J., Tsai, Y.-W., Lung, S.-C. C., Chou, C. C. K., and Chan, C.-Y.: Size-Resolved Anhydrosugar Composition in Smoke Aerosol from Controlled Field Burning of Rice Straw, Aerosol Science and Technology, 43, 662-672, 10.1080/02786820902825113, 2009.

Engling, G., He, J., Betha, R., and Balasubramanian, R.: Assessing the regional impact of indonesian biomass burning emissions based on organic molecular tracers and chemical mass balance modeling, Atmos. Chem. Phys., 14, 8043-8054, 10.5194/acp-14-8043-2014, 2014.

Fan, Y., Liu, C.-Q., Li, L., Ren, L., Ren, H., Zhang, Z., Li, Q., Wang, S., Hu, W., Deng, J., Wu, L., Zhong, S., Zhao, Y., Pavuluri, C. M., Li, X., Pan, X., Sun, Y., Wang, Z., Kawamura, K., Shi, Z., and Fu, P.: Large contributions of biogenic and anthropogenic sources to fine organic aerosols in Tianjin, North China, Atmospheric Chemistry and Physics, 20, 117-137, 10.5194/acp-20-117-2020, 2020.

Song, J., Li, M., Jiang, B., Wei, S., Fan, X., and Peng, P.: Molecular Characterization of Water-Soluble Humic like Substances in Smoke Particles Emitted from Combustion of Biomass Materials and Coal Using Ultrahigh-Resolution Electrospray Ionization Fourier Transform Ion Cyclotron Resonance Mass Spectrometry, Environ Sci Technol, 52, 2575-2585, 10.1021/acs.est.7b06126, 2018.

Sun, Y. L., Wang, Z. F., Fu, P. Q., Yang, T., Jiang, Q., Dong, H. B., Li, J., and Jia, J. J.: Aerosol composition, sources and processes during wintertime in Beijing, China, Atmos. Chem. Phys., 13, 4577-4592, 10.5194/acp-13-4577-2013, 2013.

Surratt, J. D., Gómez-González, Y., Chan, A. W. H., Vermeylen, R., Shahgholi, M., Kleindienst, T. E., Edney, E. O., Offenberg, J. H., Lewandowski, M., Jaoui, M., Maenhaut, W., Claeys, M., Flagan, R. C., and Seinfeld, J. H.: Organosulfate Formation in Biogenic Secondary Organic Aerosol, The Journal of Physical Chemistry A, 112, 8345-8378, 10.1021/jp802310p, 2008.

Xie, Q., Su, S., Chen, S., Xu, Y., Cao, D., Chen, J., Ren, L., Yue, S., Zhao, W., Sun, Y., Wang, Z., Tong, H., Su, H., Cheng, Y., Kawamura, K., Jiang, G., Liu, C. Q., and Fu, P.: Molecular characterization of firework-related urban aerosols using Fourier transform ion cyclotron resonance mass spectrometry, Atmos. Chem. Phys., 20, 6803-6820, 10.5194/acp-20-6803-2020, 2020.

Zhu, Y., Yang, L., Chen, J., Wang, X., Xue, L., Sui, X., Wen, L., Xu, C., Yao, L., Zhang, J., Shao, M., Lu, S., and Wang, W.: Characteristics of ambient volatile organic compounds and the influence of biomass burning at a rural site in Northern China during summer 2013, Atmospheric Environment, 124, 156-165, https://doi.org/10.1016/j.atmosenv.2015.08.097, 2016.

---

## Author Comment (AC2)

**Responses to Reviewer #2**

General comments.

Although lots of data have been analyzed and discussed in the manuscript, however, some additional MS/MS experiments and Kendrick mass defect analysis are encouraged. Please refer specific comments in the attached PDF.

Re: Thanks for the comments. All your suggestions are of great guiding significance to us. We revised the manuscript according to your suggestions as follows.

1. The Basically, authors have discussed a lot regarding the homologous series. I think the Kendrick mass defect should be more suitable to this purpose, particularly for twodimensional Kendrick mass defect plots such as KMD\_CH2 versus KMD\_SO3.

Re: Thanks for the good suggestion. We have added the KMD plots (Figure S6) (page 12, supplemental information). The *x*-axis direction indicates hydroxylation of the compounds, with different amounts of hydroxyl; The *y*-axis direction represents methylation, which varies in quantity and length of carbon chains.

Figure S6. Two-dimensional Kendrick mass defect (KMD) matrix plot for CHO, CHON, CHOS and CHONS compounds of High SOA-loading-D sample. The KMD [O] denotes the

Kendrick mass defect of hydroxyl functional group (-OH). The KMD [CH2] denotes the Kendrick mass defect of methylene group (CH2). The magnified star symbols represent  $C_{16}H_{24}O_8$ ,  $C_{15}H_{25}NO_8$ ,  $C_{15}H_{24}O_7S$  and  $C_{10}H_{17}NO_9S$  with relative high abundance, respectively.

**2. Authors should confirm this statement using MS/MS.**

Re: According to previous studies, the ionization capability of nitro (-NO2) and organonirate (-ONO2) is strong in the ESI(-) mode (Song et al., 2018). Some CHON compounds with relative high abundance, such as  $C_7H_7O_4N$  and  $C_7H_5O_5N$ , have been identified as typical markers of biomass burning with nitro (-NO2) and organonirate (-ONO2) structure in previous studies (Laskin et al., 2015). Thus, it is inferred that the CHON compounds in this study contain nitro (-NO2) and organonirate (-ONO2) structure. Of course, further confirmation with MS/MS does provide more evidence. This part of experiment will be made up in further research.

**3. what does 'ture' mean? (line15 page8)**

**Re: Thanks for the good suggestion. We have modified this sentence (lines 14-16, page 8).**

"The intensity-contribution of daytime samples is 23.9-25.4% higher than that of corresponding nighttime samples at moderate ones (Figure 2a-d), while it was opposite under the low SOA-loading, with a 3% higher intensity-contribution at night (Figure 2e, f)."

**4. Authors should confirme this argument with MS/MS.**

Re: According to previous studies, the ionization capability of sulfate group ( $-OSO_3H$ ) is strong in the ESI(-) mode (Xie et al., 2020). CHOS compounds with  $O/S \ge 4$  are the peaks with relative high abundance in each mass spectrum. It can be inferred that these S-containing compounds contain sulfate group ( $-OSO_3H$ ). Moreover, previous studies have shown that sulfur-containing organic matter formed by oxidation of biogenic VOCs are mainly organosulfates, and MS/MS results also show that it contains sulfate functional group (Surratt et al., 2008).

Of course, further confirmation with MS/MS does provide more evidence. This part of experiment will be made up in further research.

5. ESI(-) (line 15 page 8).

Re: Thanks for the good suggestion. We have modified this word (line 23, page 8). "ESI(-)"

**References:**

- Laskin, A., Laskin, J., and Nizkorodov, S. A.: Chemistry of atmospheric brown carbon, Chem Rev, 115, 4335-4382, 10.1021/cr5006167, 2015.
- Song, J., Li, M., Jiang, B., Wei, S., Fan, X., and Peng, P.: Molecular Characterization of Water-Soluble Humic like Substances in Smoke Particles Emitted from Combustion of Biomass Materials and Coal Using Ultrahigh-Resolution Electrospray Ionization Fourier Transform Ion Cyclotron Resonance Mass Spectrometry, Environ Sci Technol, 52, 2575-2585, 10.1021/acs.est.7b06126, 2018.
- Surratt, J. D., Gómez-González, Y., Chan, A. W. H., Vermeylen, R., Shahgholi, M., Kleindienst, T. E., Edney, E. O., Offenberg, J. H., Lewandowski, M., Jaoui, M., Maenhaut, W., Claeys, M., Flagan, R. C., and Seinfeld, J. H.: Organosulfate Formation in Biogenic Secondary Organic Aerosol, The Journal of Physical Chemistry A, 112, 8345-8378, 10.1021/jp802310p, 2008.
- Xie, Q., Li, Y., Yue, S., Su, S., Cao, D., Xu, Y., Chen, J., Tong, H., Su, H., Cheng, Y., Zhao, W., Hu, W., Wang, Z., Yang, T., Pan, X., Sun, Y., Wang, Z., Liu, C.-Q., Kawamura, K., and Fu, P.: Increase of High Molecular Weight Organosulfate with Intensifying Urban Air Pollution in the Megacity Beijing, Journal of Geophysical Research Atmospheres, 125, 10.1029/2019JD032200, 2020.